# Preparation of Cast Metallic Foams with Irregular and Regular Inner Structure

**DOI:** 10.3390/ma14226989

**Published:** 2021-11-18

**Authors:** Ivana Kroupová, Martina Gawronová, Petr Lichý, Václav Merta, Filip Radkovský, Kamila Janovská, Isabel Nguyenová, Jaroslav Beňo, Tomáš Obzina, Iveta Vasková, Ivo Lána, Jiří Rygel

**Affiliations:** 1Department of Metallurgy and Foundry, Faculty of Materials Science and Technology, VSB—Technical University of Ostrava, 708 00 Ostrava-Poruba, Czech Republic; ivana.kroupova@vsb.cz (I.K.); martina.gawronova@vsb.cz (M.G.); petr.lichy@vsb.cz (P.L.); filip.radkovsky@vsb.cz (F.R.); kamila.janovska@vsb.cz (K.J.); jaroslav.beno@vsb.cz (J.B.); tomas.obzina@vsb.cz (T.O.); jiri.rygel@vsb.cz (J.R.); 2Brembo Czech, s.r.o., 720 00 Ostrava-Hrabová, Czech Republic; Isabel_Nguyenova@cz.brembo.com; 3Institute of Metallurgy, Faculty of Materials, Metallurgy and Recycling, Technical University of Košice, 042 00 Košice, Slovakia; iveta.vaskova@tuke.sk; 4SaM Nové Ransko, s.r.o., 582 63 Ždírec nad Doubravou, Czech Republic; lana.i@slevarna.cz

**Keywords:** foundry, casting, metallic foam, inner structure, precursor, evaporative pattern

## Abstract

The aim of this paper is to summarize the possibilities of foundry methods for the production of metallic foams. At present, there are a number of production technologies for this interesting material, to which increasing attention has been paid in recent years. What is unique about metallic foams is the combination of their physical and mechanical properties. As part of our research, we designed and verified four main methods of metallic foam production by the foundry technology, whose products are metallic foam castings with regular and irregular arrangements of internal cavities. All these methods use materials and processes commonly used in conventional foundry technologies. The main idea of the research is to highlight such technologies for the production of metallic foams that could be provided by manufacturing companies without the need to introduce changes in production. Moreover, foundry methods for the production of metallic foams have the unique advantage of being able to produce even complex shaped parts and can thus be competitive compared to today’s established technologies, the output of which is usually only a semi-finished product for further processing. This fact was the main motivation for the research.

## 1. Introduction

It is a well-known fact that in nature we can commonly find materials that show interesting properties due to their internal cellular (porous) structure. Typical natural cellular materials include wood, coral or bone; even with their low weight, these materials are able to withstand high mechanical stress [1,2,3]. Highly porous materials with a cellular structure have an interesting combination of physical and mechanical properties, such as high stiffness at low weight or high permeability at high thermal conductivity [1]. Under mechanical loads, the transmitted stresses are optimally distributed in the structure that is made of them, and so there is no unnecessarily excessive stress on the structure by its own weight [4].

Currently, the most important man-made porous material is polymeric foam, which has found application in a number of industries [1]. However, porous structures can also be obtained from metals and alloys, whose properties enable the significant expansion of the field of possible applications [5].

### 1.1. Properties of Metallic Foams

Metallic foams are materials with a complex internal structure and unique properties. The internal structure of these materials consists of pores that are artificially created in the metal matrix. It is these pores, or rather their size, shape and distribution in the metal matrix that give these materials their specific combination of physical and mechanical properties [6]. Within the term “metallic foams” we can speak of materials with open and interconnected or, on the contrary, closed pores [7]. Furthermore, foam materials with or without a solid surface crust can be achieved. These parameters are already determined by the chosen production technology. In addition, all these parameters have an important influence on the final properties of the metallic foams and thus on their possible application [8,9].

The most important advantages of metallic foams include low specific density [3,10] (possibility of reducing the weight of the structure), the ability to absorb impact energy [11,12] (e.g., in the deformation zones of vehicles) and the damping ability [13] (acoustic barriers).

### 1.2. History of the Use of Metallic Foams

However, the use of these porous metallic materials in the automotive industry, construction and other areas of human activity is nothing new. The history of the use of metallic foams for engineering purposes dates back to the beginning of the 20th century. In the 1920s, metallic foam production processes based on sintering metal powders were developed. At that time, metal foams were already being used commercially for the manufacture of batteries, filters or self-lubricating bearings. The first mention of metallic foams produced by foamed melts was found in a French patent from 1925 and thirty years later in the USA it began with commercial use. However, extensive research and development in the field of metallic foam production technologies and the possibilities of their use did not begin until the 1990s and continue to this day [14].

The current trend in the development of modern construction materials is to find a suitable combination of their low specific weight and sufficient strength. A number of materials achieve these properties, including some foundry alloys [15]. However, in the field of foundry, the use of existing technological processes is a limiting factor in order to achieve thin-walled components that would also meet the requirement of low weight. The use of metallic materials with artificially created pores in the structure, i.e., metallic foams, is another possible way to reduce the weight of manufactured components without a negative impact on their mechanical properties [3].

As already indicated, metallic foams are currently used in a wide range of areas. In the construction industry they can be used, for example, as protective elements in fire protection [16]. In engineering, their low weight is used to lighten structures [17]. In transport, metallic foams are successfully used in vehicle crumple zones due to their ability to absorb impact energy [18]. Metallic foams are also used in various applications due to their excellent acoustic properties (noise absorption) [19]. A very interesting area for the potential application of metallic foams is medicine, where metallic foams made of special alloys may find application as bone implants due to their cellular internal structure [20,21].

In addition to functional applications resulting from the specific properties of metallic foams, these materials have been used in recent years in the field of design and art for their unconventional and attractive appearance [22].

### 1.3. Metallic Foams Production Options

Since the discovery of porous metals, a number of methods have been developed for their production. Some technologies are similar to polymer foaming processes, others are developed with regard to the characteristic properties of metallic materials, such as their ability to sinter or the fact that they can also be applied electrolytically [4].

The porosity is most often in the range of from 30 to 97% [4], depending on the production method and the used material. By changing the process parameters, it is possible to obtain a porous structure with different sizes and shapes of pores and different types of their arrangement, either irregular or regular [4,23].

Production processes can be divided into four main groups, which differ from each other by the use of the starting material, or the state in which this material is in [24,25]. Porous metal materials can be made of metal vapor [26], liquid metal [27], powdered metal [28,29] or metal ions [4].

It is the production technology of this unique material that is in many cases the limiting factor for its use in common applications. The technology for preparing metallic foams is in most cases quite complex and requires special equipment and procedures. Despite the fact that we are currently encountering applications of these materials, these are only very rare cases and the full use of the application potential of metallic foams is still not used [4,22].

### 1.4. Current State of Metallic Foam Production in the Czech Republic

Current research in the field of development of production processes for porous metals is therefore generally focused on finding undemanding technologies for their production. One possible way is to introduce conventional foundry processes into metallic foam production technology. Therefore, within the research attention is paid to the technologies of preparation of metallic foams and porous metals from the liquid phase. Foundry technologies offer an economical, time-saving and in some cases ecological possibility of producing shape castings of metallic foams with a wide range of internal structure morphology in connection with the shape, size, regularity of distribution or degree of interconnection of internal pores. At present, some of the foundry methods for the production of metallic foams in the world are already in commercial use [30,31,32]. However, that is why it is necessary to pay attention to these technologies and further develop them: there are over 200 foundry plants in the Czech Republic, but none of them deals with the production of metallic foams.

## 2. Materials and Methods

Our designed and proven technologies for the production of cast metallic foams are based on common foundry processes and materials. These methods can be divided into two basic groups according to the regularity of the internal cavities’ arrangement of the final casting into:technology of production of cast metallic foams with irregular arrangement of internal cavities [33,34];technology of production of cast metallic foams with regular arrangement of internal cavities [35,36].

For each type of material, two production technologies were designed and verified (Figure 1). For each of the technologies, the individual technological steps and the materials used were verified. An overview of the individual technologies is shown in Table 1. The aim was to verify the technology of metallic foam production by casting into conventional foundry moulds on the basis of laboratory and semi-industrial tests. These processes allow the production of shaped castings with pores, using equipment common in foundries with traditional technology.

In addition, the material of the test castings itself was chosen with regard to the commonly used castings materials. Thus, the technologies were verified for the Al-Si type alloy (ČSN EN 1706), the Cu-Sn type (EN CC483K) and for the graphitizing iron alloy (EN GJL-200).

The following production technologies were designed and verified within the technology of the production of cast metallic foams with irregular arrangement of internal cavities:infiltration of metal into the cavity of the mould filled with precursors;two-stages investment casting process using a polyurethane evaporable pattern.

Within the technology of production of cast metallic foams with a regular arrangement of internal cavities, the following production technologies were designed and verified:infiltration of metal into the cavity of the mould filled with the preform;technology using a disposable evaporable pattern.

### 2.1. Infiltration of Liquid Metal into the Cavity Filled with Precursors

The first proposed and verified technology for the production of metallic foams with an irregular structure by the foundry process is a method using precursors which fill the cavity of the foundry mould. The term precursor in this paper refers to a particle (preferably spherical in shape) made most commonly from common foundry core mixtures. These particles help to create pores in the volume of the casting after pouring with liquid metal: after solidification of the metal they are removed from the casting and thus form the resulting cavities. In order to ensure convenient removal of the filler material, it is desired to produce precursors with excellent collapsibility upon thermal exposure. At the same time, a sufficient number of precursors in the mould cavity must also be provided in order to interconnect the individual pores. If all precursors did not come into contact with each other, they could remain encapsulated in the volume of the casting and a composite material (so-called syntactic foam or bimodal composite metallic foams) would be formed [37,38]. Precursors can be made of various materials. In general, this filling material is required to have sufficient heat resistance and subsequent good collapsibility after casting.

#### 2.1.1. Material and Precursors Production

Within the experimental work, several types of precursors were tested, regarding the materials from which this filler material was made. These granules also differed from each other in used production technology. Examples of individual types of precursors can be seen in Figure 2.

##### Ceramic Precursors

Although it is a material that is commercially produced and is therefore commonly available, the results of the experiment do not predict a wider use of ceramic precursors in the production technology of cast metallic foams. The ceramic material has not proven to be very suitable for the given purpose, due to its low density its arrangement in the structure of the solidified casting is too affected by the dynamic effect of the flowing molten metal.

However, one of the possible areas of use of this material would be the area of production of so-called syntactic (or bimodal composite) metallic foams, i.e., a special type of composite material, where the precursors remain "encapsulated" in the metallic material. It then acts in the given structure as a lightening and further to improve some required mechanical properties.

##### Precursors from Cores Produced by Croning Technology

This material proved to be suitable for the possibility of production of filling material—precursors—for the evaluated technology of production of cast metallic foams. The main advantage of this technology is the fact that mixtures of new raw materials are not prepared for the production of these precursors, but waste non-conforming cores are used. The use of these cores for the production of precursors has thus resulted in the recycling of materials which, under normal conditions, would have to be disposed as hazardous waste. It is therefore possible to assume further use of the material, which would be in other cases a burden for the foundry operation.

The disadvantage of these precursors is their irregular shape, which is caused by the uneven tumbling of the individual fragments due to the irregular hardening of the core in its cross section.

##### Furan No-Bake Mixture

As in the previous case of using a commonly available core mixture, in this case the core mixture has proved its worth and can be evaluated as a suitable material for the production of precursors. In the case of the furan ST mixture, it is prepared from new raw materials. However, in the case of technology implementation into practice, the possibility of using the so-called "residual" core mixture can be considered. It is a material that normally accompanies the production of cores at a foundry plant, and if it is not used, it becomes waste for the foundry. This fact is therefore a major advantage of this technology. The precursor material also showed very good collapsability after casting, which is a very important aspect in this technology.

The main disadvantage of this procedure could be the non-reproducibility of the results and the fact that each casting is unique and therefore its resulting mechanical properties can not be completely predicted. However, this disadvantage is common to a whole group of metallic porous materials with an irregular distribution of internal cavities.

##### PUR Cold Box

This type of precursor is unique in that it uses the second branch of the development of cast metallic foam production technologies at the Department of Metallurgy and Foundry, namely the production of metallic foams with a regular internal structure by liquid metal infiltration into a mould cavity filled with a preform with precisely defined geometry. For the purposes of the production of these materials, the complex cores are made in the operating conditions of SaM Nové Ransko, s.r.o., which are inserted in the levels into the cavity of the foundry mould, where they form the so-called preform. However, during the production and subsequent handling of such complex cores, they may be damaged, and it is these discarded cores that have been used to make precursors of the same size (ø 10 mm) and shape to produce metallic foams with irregular arrangements of internal cavities.

It is the regular shape and the same size of the precursors that represent the main advantage of this technology. In addition, the possibility of using waste material—discarded cores made for the production of castings with a regular arrangement of internal cavities—can be considered as a positive of this process.

The disadvantage is the reduced collapsibility of the precursors after casting: due to the fact that the precursors fill a large part of the volume of the mould cavity, the mould: metal ratio is unfavorable, and the precursors must be removed mechanically from the casting volume.

##### Salt Precursors

The use of salt precursors seems to be very suitable from the point of view of the possibility of their removal from the volume of the casting. After casting, these can be removed by washing from all internal cavities immediately after the casting has solidified (on condition that the precursors have touched each other to form interconnected cavities). This eliminates the complicated process of the mechanical removal of the filling material, which was a common disadvantage of all the above-mentioned processes.

I recommend paying attention to the use of a salt-based core mixture for the possibility of making precursors or preforms (regular distribution of the internal cavities of the metallic foam) in the following experiments. The mentioned material appears to be very suitable for the given application precisely due to the undemanding process of removing (washing out) the filling material from the volume of the casting.

#### 2.1.2. Placing of Precursors into the Mould Cavity

The technology of infiltration of liquid metal into the cavity of a mould filled with precursors fully met the requirement of undemanding technology using conventional foundry processes and materials. However, its main disadvantage was the process of placing the precursors into the mould cavity. It was very simple in the case where the whole cavity was stored only in one (lower) half of the mould in the case of two-frame moulding (Figure 3). In this case, the precursors were loosely poured into the cavity located in the lower half of the mould and then "weight down" with the full upper half of the mould.

However, the whole process was more complicated in the case of the need to make a more complex shape of the casting, in which case the geometry did not allow us to store the pattern (and create a cavity) only in one half of the mould. For this situation, it was necessary to devise measures for securing the poured precursors in the mould cavity at the moment of tilting the upper mould frame and folding the mould (Figure 4).

From the point of view of the method of storing the filling material in the mould cavity, several possible methods were verified (polystyrene and paper semi-finished products inserted into the mould cavity, fixing of precursors in the upper frame by a strip foil). The best method for securing the filling material in the mould was to use a strip foil. The bulk filling material was firmly held in the upper part of the mould when the mould was assembled. This undemanding method can therefore also be applied in operating conditions.

#### 2.1.3. Foundry Mould

The mould for making metallic foam by the technology of infiltration of liquid metal into the cavity filled with precursors can be made from any commonly used materials for the foundry sand mould.

In our experiments, each mould consisted standardly of two frames, which intentionally formed a mould joint, which was important for tests of placing precursors in the formed mould cavity. A green sand mixure was first used in the moulding, and its composition was:Basic sand—silica sand—100 wt%;Binder—bentonite—7 wt%;Water—3.7 wt%.

After making several samples, Dextrin was intentionally added to this green sand mixture to ensure the mixture’s greater plasticity and to improve the drying resistance. Dextrin was added to the green sand mixture of 0.5 wt%.

#### 2.1.4. Casting Process

All test castings were made by conventional gravity casting technology into disposable foundry moulds from the above moulding compound. The material of the test castings were common foundry alloys based on Al, Cu and Fe. The melting of the metal took place in an electric resistance furnace (LAC, s.r.o., Židlochovice, Czech Republic) or in an electric induction furnace (INDUCTOTHERM GROUP, Rancocas, NJ, USA) (depending on the cast material).

#### 2.1.5. Cleaning of Castings

As previously indicated, the precursors did not need to be removed from the casting volume after the casting process. In this case, we are talking about a material of the syntactic foam type. Normally, however, precursors are removed from the volume of the casting after casting process, solidification and cooling: pores are formed in the volume of the casting. The removal of precursors from the volume of the casting takes place mechanically and these procedures include the usual cleaning work used in foundry operations (knocking, vibration, etc.).

### 2.2. Two-Stage Investment Casting Process Using a Polyurethane Evaporable Pattern

The second designed and verified technology for the production of metallic foams with an irregular structure by the foundry process is a method using an evaporable pattern of polymer foam. The principle of the method consists of pouring the polymer foam with a refractory material (Figure 5). The mould thus created is then dried and annealed: at high temperatures the polymer model is vaporised. This creates a cavity in the mould into which the liquid metal is cast. After the metal has solidified in the mould and the moulding material has been removed, we obtain a metallic foam that is an exact copy of the original polymer pattern.

The appropriate selection of moulding material is crucial in this technology. The main requirements for the material are good flowability (to be able to fill the entire mould and all the complex internal cavities of the polymer pattern) and refractoriness. In our experiments, we worked with a mixture based on calcium sulphate hemihydrate.

#### 2.2.1. Material and Pattern Modification

At the beginning of the casting production process is the selection of a suitable pattern type/pattern material. The proper choice of pattern/pattern material fundamentally affects the quality of the final casting. In the case of two-stage investment casting technology using a disposable evaporable pattern, a polymer foam with fully open interconnected pores is used as the pattern. These parameters are met by a polymer foam from a Czech manufacturer (Eurofoam TP, s.r.o., Brno), specifically a reticulated polyurethane (PU) foam based on polyester. The individual types of foams used for the production of evaporable patterns can be seen in Table 2.

The patterns themselves for the technology of two-stage investment casting are then made by cutting the desired shape (the shape of the final casting) from a block of polymer foam (see Figure 6).

Even after selecting the appropriate type of polymer foam, the use of this material on patterns has certain pitfalls. The manufacturer of polymer foams is able to guarantee accurate porosity (cell diameter, PPI), but during the production process (reticulation) it is not possible to ensure the same thickness of the polymer foam fibers (ligaments) in its entire volume. The individual foundry patterns can thus have different fiber thicknesses, which significantly affects the fluidity of the liquid metal into the cavity, which remains in the mould after the pattern has evaporated. In this case, the fluidity can be increased by artificially increasing the thickness of the polymer foam fiber, for example, by dipping it in wax, or by spraying paint. It was experimentally verified that the most effective method of surface treatment of a polymer pattern is the procedure with the application of one layer of paint: the patterns treated in this way led to an increase in the fluidity of the metal into the complex cavity of the mould.

#### 2.2.2. Foundry Mould

For the technology of a two-stage investment casting process, it is necessary to choose a material for the mould that will be able to completely fill all the cavities of a very complex polymer pattern. At the same time, however, it must show good collapsibility after casting: the possibility of removing the moulding material from the complex structure of the resulting casting.

From the point of view of the mould, we can influence the fluidity of the liquid metal especially by the mould temperature. The higher the casting temperature the mould will have, the better the liquid metal will run into all its cavities, and there will be no heat shock and premature solidification of the metal in the mould. For this reason, the moulding material must have good heat resistance and low thermal expansion (dimensional change with temperature changes). Several types of plaster materials intended for investment casting technology were tested experimentally, and the plaster mixture marked Kerr Lab Satin Cast 20 (Kerr Corporation, Brea, CA, USA), which met all the above requirements, proved to be the best. As the manufacturer does not specify the exact mineralogical composition of the plaster supplied by him, the manufacturer’s recommendation was used to dilute the plaster.

#### 2.2.3. Casting Process

The Indutherm laboratory casting line with the designation MC15 was used for the melting and casting process itself. Thanks to a very powerful induction furnace, a wide range of metals and alloys with a melting point of up to 2000 °C can be melted in the device. The power of the inductor, which is 3.5 kW, can be regulated from 20% to 100% (individual steps of 10%). The melting itself then takes place in various types of specially shaped ceramic crucibles with a spout, which ensures the direction of the metal flow into the inlet well. For non-magnetic metals, it is suitable to use a crucible with a graphite lining.

During melting, a reduced pressure (20 mbar) is created in the working chamber. Thanks to this, the possibility of oxidation and gasification of the melt is minimized, which has a very positive effect on the quality of the cast material.

The casting itself into a preheated plaster mould located in the machine chamber under the crucible nozzle takes place by a tilting mechanism (tilting the cabin by 90°). When the entire furnace chamber is flipped over, the melt is poured out of the crucible into a plaster mould. At the moment of overturning of the chamber, the pressure equalizes, and the pressure increases immediately (overpressure up to 2 bar). This helps to run (“push”) the metal into the mould cavity and acts in the machine cabin throughout the solidification and cooling of the casting. It is also possible to set a delay time for the formation of overpressure from the moment of overturning, in particular at an interval of 0–5 s.

#### 2.2.4. Cleaning of Castings

After removing the casting from the mould, a considerable amount of used plaster mixture still remains in the pores of the cast metallic foam, which must be removed from the volume of the casting. Based on previous research, the castings were cleaned by leaching in a 1 M solution of nitric acid (HNO_3_). In this process, the insoluble calcium sulfate dihydrate CaSO_4_·2H_2_O is converted to water-soluble calcium nitrate Ca(NO_3_)_2_. Subsequently, the castings were cleaned in an ultrasonic bath (FRITSCH GmbH, Weimar, Germany). However, the process is very time consuming: for the complete cleaning of the casting, at least 1 h of leaching in HNO_3_ solution and 1 h of cleaning in an ultrasonic bath was required. At the same time, this procedure carries with it a great environmental burden in the form of the disposal of used acid.

Therefore, a suitable replacement for this process was sought. The authors [4,39] mention in their works the possibility of using pressurized water for cleaning these castings. As an alternative to this variant, the use of compressed air (pressure 10 bar) has been tried. This procedure has proven to be very effective for cleaning castings of given dimensions. The “blowing” of the plaster mixture residues from the casting took less than 5 min. For complete cleaning, an ultrasonic cleaner was used again, in which, however, it was sufficient to leave the castings for less than 15 min to achieve perfect cleaning. The disadvantage of the procedure is a slightly higher dust content.

### 2.3. Infiltration of Liquid Metal into a Cavity Filled with a Preform

The principle of the technology of the infiltration of liquid metal into the cavity of the foundry mould, which is filled with a complex core, preform (Figure 7), is similar to the technology using precursors. Unlike these, the preform has a clearly defined shape and the resulting metallic foam is characterized by regular placement of the internal cavities. In addition, it is possible to make castings with a solid wall between the individual levels of the cells: castings with such an internal structure can subsequently be used in the field of thermal engineering as heat exchangers.

#### 2.3.1. Material and Production of Preform

The preform in the mould fulfills the purpose of the filling material, as well as the precursors of the technology in Chapter 2.1. Unlike the precursors, however, the preform has a precisely defined geometry. Due to its function, the preform can be imagined as a complex core.

The production of cores took place in the operating conditions of SaM Nové Ransko s.r.o. by shooting using PUR Cold Box technology. This core production technology is based on a two-component binder system (benzyletherpolyol and diphenylmethane-4-4-diisocyanate) and represents a suitable process for the production of such a geometrically and shape-complex preform. The mixture was prepared above the machine in a vertical paddle mixer (with four straight scraper blades), transported by gravity over the core box and then shot into it on a LAEMPE core machine (Laempe Mössner Sinto GmbH, Barleben, Germany). Curing was accomplished by blowing the core with a gaseous tertiary amine C_6_H_15_N (ASK Chemicals CZ s.r.o., Brno, Czech Republic) which acts as a catalyst to form a polyurethane resin. It is therefore a cold curing process. The resulting preform composed of five layers of cores can be seen in Figure 8.

#### 2.3.2. Foundry Mould

The same type of moulding compound was chosen for these experiments as for the precursor technology. Within the technology with the inserting of the preform, each form consisted standardly of two frames, and the composition of green sand mixture was:Basic sand—silica sand—100 wt%;Binder—bentonite—7 wt%;Water—3.7 wt%.

#### 2.3.3. Casting Process

The test castings were made by gravity casting technology into disposable foundry moulds from the above-mentioned molding mixture. The material of the test castings were common foundry alloys based on Al, Cu and Fe. The melting of the metal took place in an electric resistance furnace or in an electric induction furnace (depending on the material to be cast).

#### 2.3.4. Cleaning of Castings

As with the previous type of filling material, it would not be necessary to remove the preform from the volume of the casting after the casting process: we would then discuss so-called syntactic foams. However, our intention was to create a metal material with a regular distribution of internal cavities, so we removed the filling material of the preform after the casting and cooling process from the casting mechanically by shaking off with the help of vibrations.

### 2.4. Technology Using a Disposable Evaporable Pattern

The method basis is an evaporable pattern with a complex geometry, which is provided with a protective coating, and is left in the mould during casting. The pattern is made of expanded polystyrene (EPS), assembled including the gating system by cutting, gluing and grinding. Dry sand without added binders can be used as moulding material. The biggest advantage of this technology is the simple composition of the moulding mixture and the fact that the pattern is not removed from the mould and can thus have almost any complex construction. The principle of the technology is shown schematically in Figure 9.

#### 2.4.1. Material and Pattern Modification

As already mentioned, as a material for the production of the pattern (including the gating system) this technology uses a material that evaporates during the casting of liquid metal, most often EPS, i.e., plastic material, which has special properties due to its structure. EPS is composed of individual low-density polystyrene cells and is extremely light.

According to the design obtained from the previous research, patterns of grids, including the gating system, were produced. The grids were glued from supplied semi-finished products from the company H-TIPOL Halašta, spol. s r. o. (Zašová, Czech Republic). To create the whole pattern, seven individual grids were used, which were glued to each other. One level—the grid—had dimensions of 52 × 52 × 6 mm and the total size of the pattern was 52 × 52 × 42 mm. A 3D model of the whole assembly, i.e., the gating system and two patterns, can be seen in Figure 10.

The quality of the application of the protective coating for the patterns plays a very important role in this technology. The protective coating must be strong enough to maintain the pressure of the sand as the pattern evaporates during casting and at the same time be sufficiently permeable to allow to pass the gasses from the patterns’ space into the mould. It should also not worsen the surface of the casting and possibly adversely affect the size of the casting. The alcohol coating Foundrylac ZBM/365 (Mazzon, Vicenza, Italy) was used. It is often used as an affordable replacement for zirconium coatings and is suitable for cast iron castings, but has also been verified for aluminum alloys. The coating was thoroughly mixed, and the viscosity control was performed on a viscosity cup. The outflow time was supposed to reach 18 s. The pattern of the porous metal was complicated, so the method of coat soaking was chosen. After 24 h of drying at room temperature, the inspection took place, and the necessary repairs were made with a brush or the pattern was completely soaked and provided with a second layer of coating and dried again until the next day (24 h).

#### 2.4.2. Foundry Mould

In the case of technology using an evaporable pattern, the moulding mixture usually consists only of sand, in our case, it was silica sand Grudzeń Las (Grudzeń Las. Sp. z o.o., Grudzeń Las, Poland) with the designation GL 21 (d_50_ = 0.20–0.24 mm). This sand is characterized by an oval grain shape, a smooth surface and high purity (SiO_2_ content over 99.2%). This sand was pre-dried in an oven at 105 °C before use and moulding was performed in a special moulding frame with a bottom. Dimensions of the moulding frame were l × w × h = 340 × 230 × 160 mm. At the bottom of the mould was sand first poured loosely into the height of a few centimeters. The polystyrene evaporable pattern with the gating system provided with paint was placed on the sand and covered with sand up to the edge of the grid pattern. The mould was then vibrated for 90 s to compact the sand. In the next step, the sand was poured up to the edge of the mould and the whole mould was vibrated again for 180 s. In the case of other moulds produced in this way, the surroundings of the sprue base were reinforced by the green sand mixture. The reinforcement retained the edge of the sprue base and there was no failure due to inaccuracy during casting.

#### 2.4.3. Casting Process

The material for casting was aluminum alloy AlSi_10_MgMn, which is suitable for the production of thin-walled and complex castings. The alloy was melted in an electric resistance furnace (LAC 80/13). After removing the crucible with molten metallic from the furnace, the oxide membrane was removed from the surface of the melt and the temperature of the metal was measured using a thermocouple (Omega Engineering Inc., Norwalk, CT, USA).

At a pouring temperature of 730 to 739 °C, all castings were of good external quality, the melt perfectly filled the mould and replaced polystyrene. The protective coating kept its shape even after the casting process and could be easily removed by knocking. The pouring temperature of 720 °C was tested in the experiments. However, it was not enough, because the metal did not run into the upper parts of the casting. The coating showed sufficient strength and kept its shape after being pulled out of the mould, even in the case of a scrap casting.

#### 2.4.4. Cleaning of Castings

Due to the nature of the used moulding mixture (dry sand without binder), the cleaning operations in this case are considerably simpler than with previous technologies. After removing the finished casting from the mould, it was necessary to remove the residues of the thermally degraded protective coating. The casting itself was cleaned by shaking and using a steel brush. Subsequently, the gating system was removed.

## 3. Results

The above text shows that to verify the possibility of making metallic foams by foundry technology the two main procedures were designed by casting over filler material (precursors or preforms) placed in the mould cavity and using a disposable evaporable pattern to create a complex metallic foam structure.

### 3.1. Results of Technology Using Precursors

The results of the proposed technology are castings of metallic foams (Figure 11) with a porosity up to 70%. Porosity—this essential property of metallic foam material—can be influenced by the amount, size and shape of the filler material (precursors).

Castings using this technology can be made from commonly used foundry alloys based on Al, Cu and Fe.

Due to the complexity of the resulting castings and the high shape and dimensional variability, these products can be applied, for example, in the field of filtration or as a construction material.

### 3.2. Results of the Technology of a Two-Stage Investment Casting Process

This process of the production of cast metallic foams is the most demanding of the presented procedures both in terms of time and economy. However, its biggest advantage still remains the possibility of producing metallic foams with the highest porosity (up to 97%). The resulting metallic foam castings can be seen in Figure 12.

Castings with this technology can be made from commonly used foundry alloys based on Al, after adjusting the annealing cycle (heating the mould to a higher temperature), and the technology is also suitable for Cu alloys.

Due to the complex internal structure of the resulting castings, where all pores are interconnected and thus allow the passage of liquid or gaseous media, these products can be applied, for example, in the field of filtration.

At present, our further attention is focused on the modification of the polymer evaporable pattern, regarding the possibility of increasing the thickness of its fiber. This technological step achieves the strengthening of the cross section of the PUR foam ribs, which has a positive effect on the resulting mechanical and selected functional properties of the metallic foam casting. Many authors worldwide are currently addressing this issue [39,41].

### 3.3. Results of the Technology Using Preform

The results of the proposed technology are metallic foams castings (Figure 13 and Figure 14) with a porosity in the range of 45 to 55%. Thanks to the specific design of the preform, the control of the size and shape of the resulting cavities at the final castings can also be controlled as well their distribution in the volume of the casting.

Castings, using this technology, can be made from commonly used foundry alloys based on Al, Cu and Fe.

Due to the regular distribution of internal cavities in the casting with the possibility of creating a solid wall between the individual cells, it is possible to use these castings with a large surface area in the field of thermal technology, i.e., heat exchangers.

### 3.4. Results of the Technology Using an Evaporable Pattern

The results of the proposed technology are castings of metallic foams with a porosity of 50%. The individual cavities are regularly distributed in the volume of the casting: their distribution is determined by the placement of individual ribs made of EPS on the pattern itself.

The resulting cast metallic foam with a regular structure (Figure 15) then meets the generally stated unique properties of metallic porous materials, i.e., low specific weight, high stiffness and strength, ability to absorb deformation energy and at the same time very good acoustic and insulating properties.

Due to the complexity of the resulting castings, these products can be applied, for example, as a construction material. In Figure 16, we can see a detail of the final casting.

## 4. Discussion

The experimental part of the presented work was focused on foundry methods for the preparation of metallic foams with irregular and regular distribution of internal cavities. Foundry technologies could offer an economical, time-saving and in many cases ecological possibility of producing metallic foams with a wide range of internal structure morphology in connection with the shape, size, regularity of distribution or degree of interconnection of internal pores. In the experiment, special attention was paid to the technology of infiltration of the liquid metal into a mould filled with filler material (precursors, preform) and technology using an evaporable pattern (two-stage process of investment casting and evaporable pattern casting). The main goal was to find a process for the production of porous metals, which is based on commonly used foundry technologies. Achieved parameters of the resulting metallic foam castings are shown in Table 3. The partial results of the used procedures are summarized in the following points:Area of infiltration of liquid metal into the cavity filled with precursors: this process for the production of cast metallic foams with an irregular distribution of internal cavities most closely corresponds to the ideas of conventional foundry technology using existing materials and technologies. The technology offers us a fast and undemanding way of producing cast metallic foams with an irregular distribution of inner cells, in which, however, we are able to define the shape, size and degree of interconnection of the inner pores. In this phase of the experiment, attention was focused mainly on testing various materials for the production of precursors.Area of the two-stage investment casting process: this process of production of cast metallic foams is the most demanding of the presented procedures both in terms of time and economy. However, its biggest advantage still remains the possibility of producing metallic foams with the highest porosity (up to 97%). The course of the experiments made it possible to determine the optimal metal fluiditity conditions and pouring conditions for this type of castings with a very small metal fiber thickness (0.19–0.40 mm). Detailed experimental conditions are given, e.g., in literature [42].Area of infiltration of liquid metal into the cavity filled with the preform: as with precursor technology, this is close to the idea of conventional foundry technology using existing materials and technologies. An additional advantage is the fact that thanks to the use of preforms with a precisely defined geometry, we are able to create metallic foams with a regular and clearly defined distribution of internal cavities. The technology of production of the preforms themselves, i.e., very complex cores, remains a question.Area of technology using a disposable evaporable pattern: this technology also enables the production of metallic foams with a regular arrangement of internal cavities. The biggest advantage of this procedure is the possibility of using a binder-free moulding mixture: the complex inner structure of the future casting is given by the construction of the pattern and it is not necessary to use a core to achieve it.

## 5. Conclusions

It has been shown that metallic foams with regular and irregular internal cavity arrangements can be produced by conventional foundry processes. Some proposed and proven technologies do not even require the use of any non-standard materials. What is unique about these processes is that they allow the production of shaped metallic foam castings, which make them competitive with existing technologies. Using the same casting principles (e.g., infiltration of liquid metal into a mould cavity filled with filler material), we are able to achieve both a non-uniform (using precursors) and uniform (using preforms) distribution of internal cavities in the casting. In addition, one of the investigated technologies allows for the use of waste material: knocked out cores after casting. This appears to be interesting in view of the increasing demands for waste-free production. It is assumed that the knowledge of the procedures, parameters and conditions of production can contribute to the expansion of the production portfolio of conventional foundry plants with a new type of product. This step can contribute to increasing the competitiveness of these production plants. At present, there is no company in the Czech Republic that is involved in the production of (any kind of) metallic foams.

The condition for the full utilization of the application potential of metallic foams is the managing of inexpensive methods of their production without additional demands on the equipped production workplaces with any capital-intensive machines and equipment. The experimental part of the presented work was devoted to this topic, namely, foundry technologies for the production of metallic foams based on conventional foundry processes.

## Figures and Tables

**Figure 1 materials-14-06989-f001:**
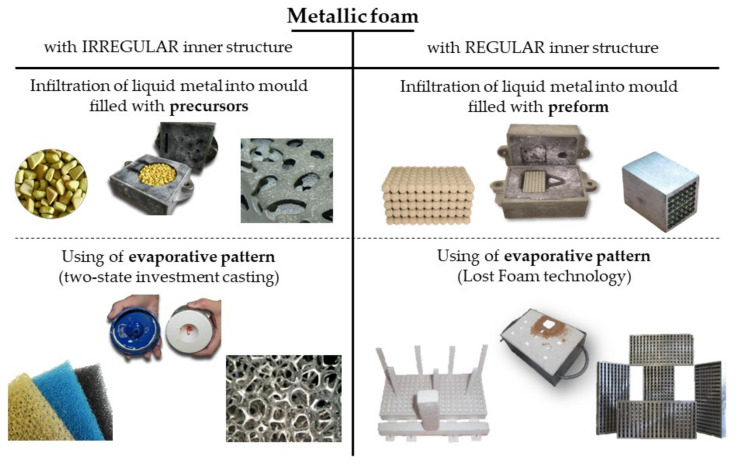
Sorting of production technologies of metallic foams by the foundry process.

**Figure 2 materials-14-06989-f002:**
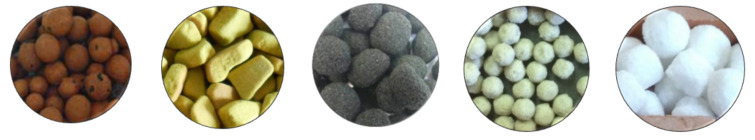
Example of individual types of precursors, from left: ceramic material based on Al_2_O_3_∙SiO_2,_ shards of cores (Croning technology), furan no-bake mixture, PUR Cold Box, salt precursors.

**Figure 3 materials-14-06989-f003:**
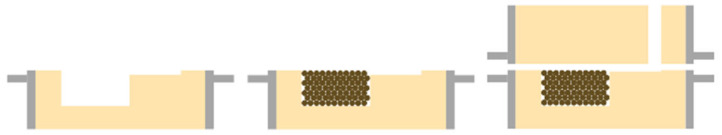
Precursors in the lower half of the mould.

**Figure 4 materials-14-06989-f004:**
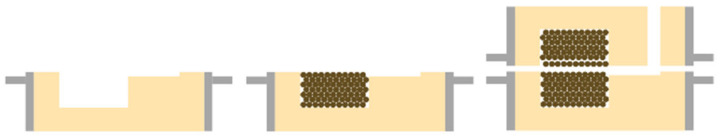
Precursors stored in both frames.

**Figure 5 materials-14-06989-f005:**
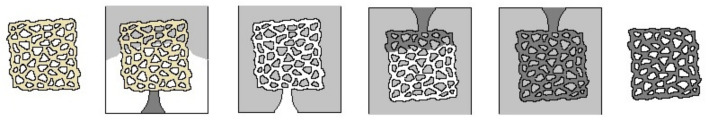
From left: polymer foam, polymer foam filled with plaster, evaporation of polymer, infiltration of liquid metal into the resulting cavity, metallic foam in the mould, final casting of metallic foam.

**Figure 6 materials-14-06989-f006:**
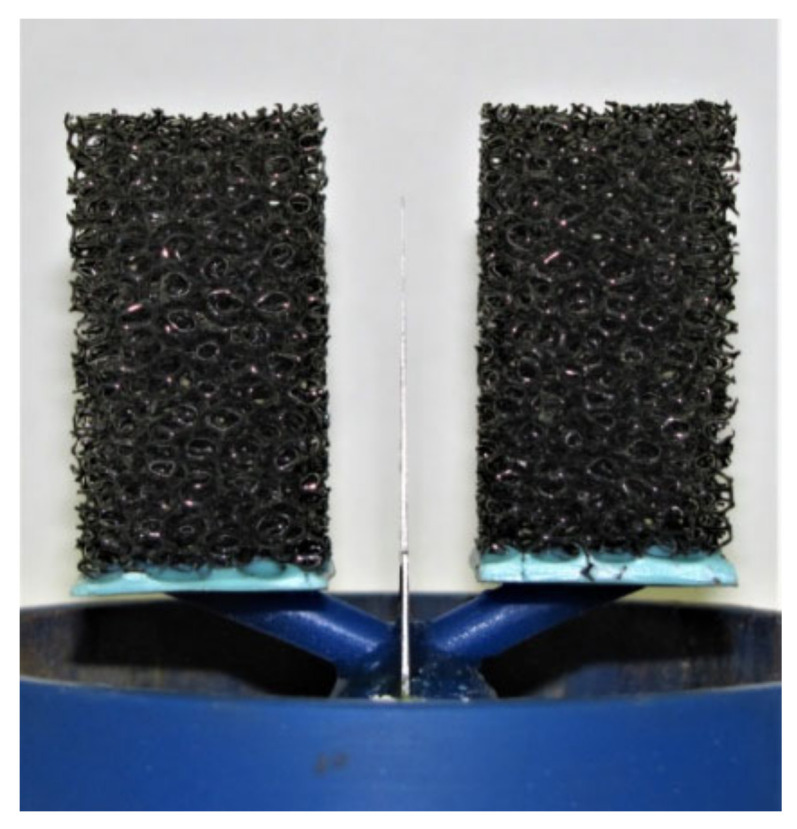
Polyurethane patterns with inlet system—full wall inlet.

**Figure 7 materials-14-06989-f007:**
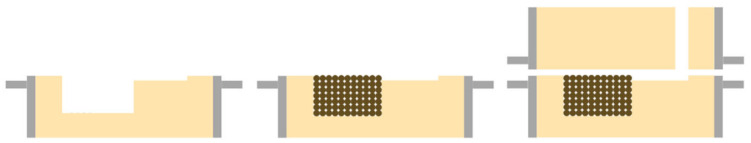
The principle of infiltration of liquid metal into the cavity of a mould filled with a preform.

**Figure 8 materials-14-06989-f008:**
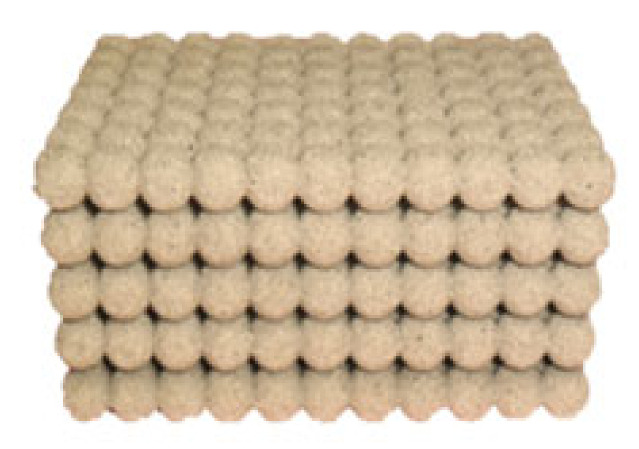
The resulting preform ready to be insert into a mould.

**Figure 9 materials-14-06989-f009:**
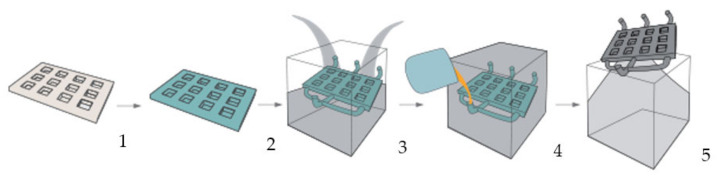
The method basis is an evaporable pattern: 1—assembly of the EPS pattern; 2—application of coating; 3—mould preparation; 4—casting of liquid metal on the evaporable pattern; 5—removal of the final casting [40].

**Figure 10 materials-14-06989-f010:**
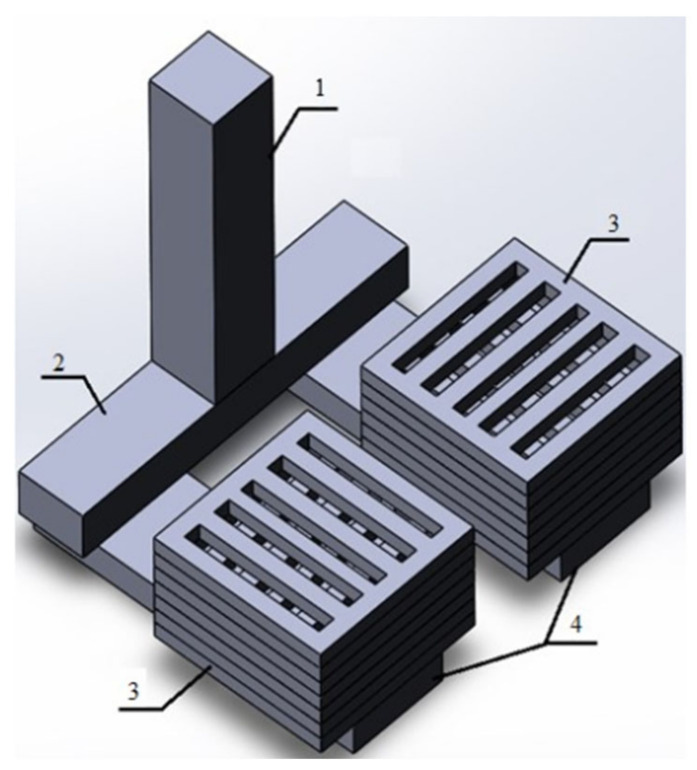
3D model of the casting: 1—sprue; 2—runner; 3—metallic foam patterns; 4—gates.

**Figure 11 materials-14-06989-f011:**
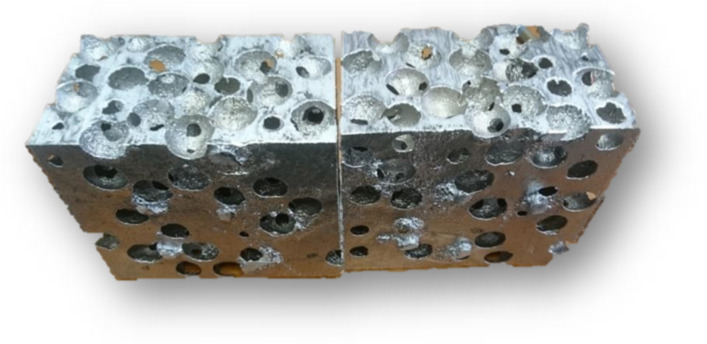
Section of the resulting casting: metallic foam with an irregular structure, AlSi_10_MgMn alloy (PUR Cold Box precursors).

**Figure 12 materials-14-06989-f012:**
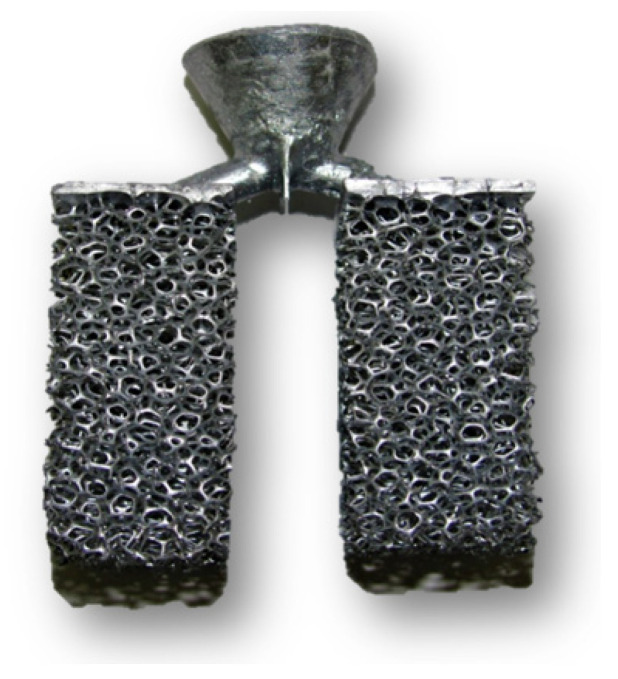
Cleaned test casting.

**Figure 13 materials-14-06989-f013:**
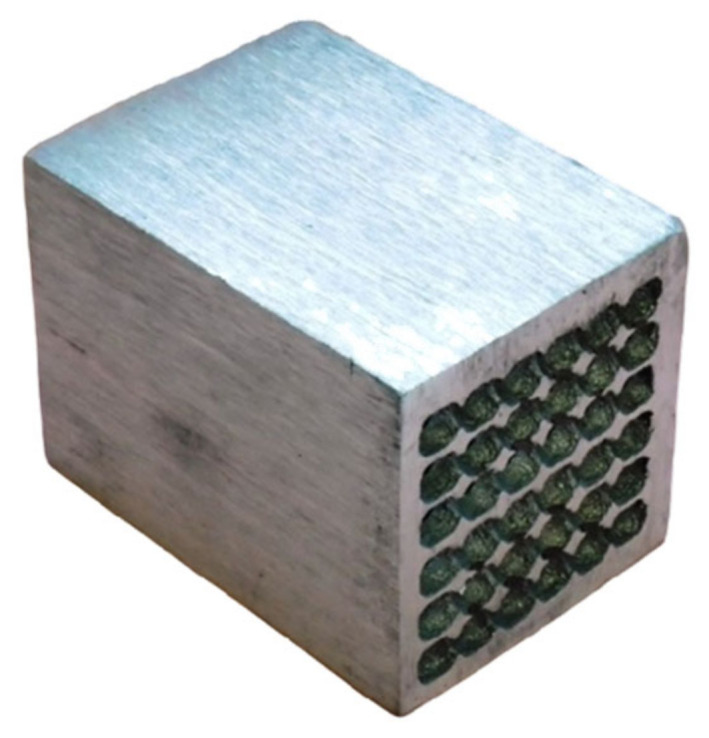
Metallic foam casting with regular arrangement of internal cavities.

**Figure 14 materials-14-06989-f014:**
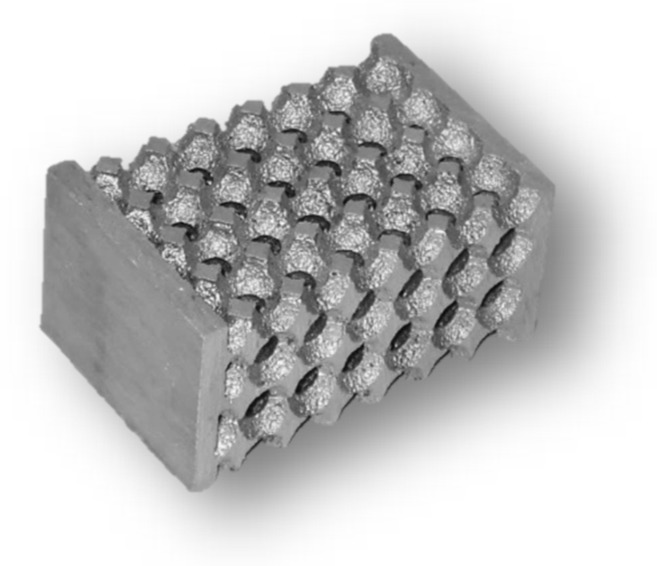
Cutout of metallic foam casting.

**Figure 15 materials-14-06989-f015:**
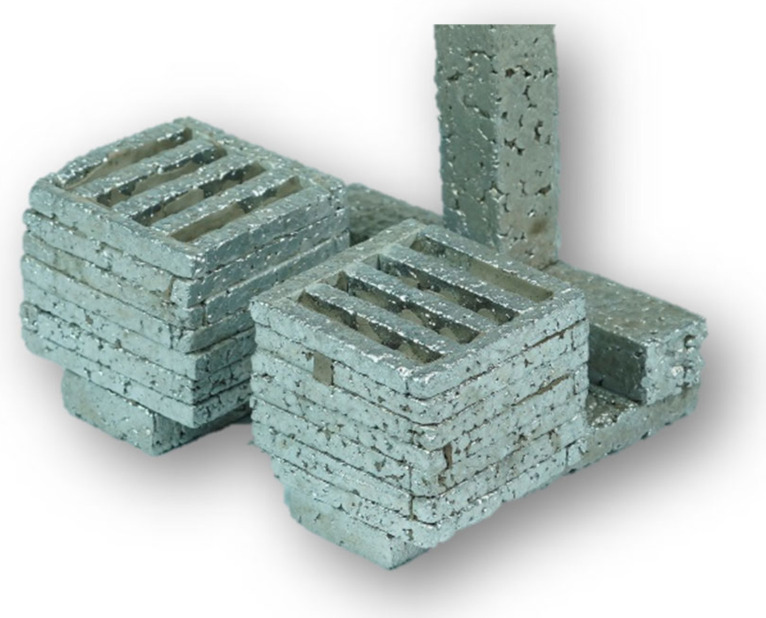
Final pair of castings with gating system after removal from the mould and cleaning.

**Figure 16 materials-14-06989-f016:**
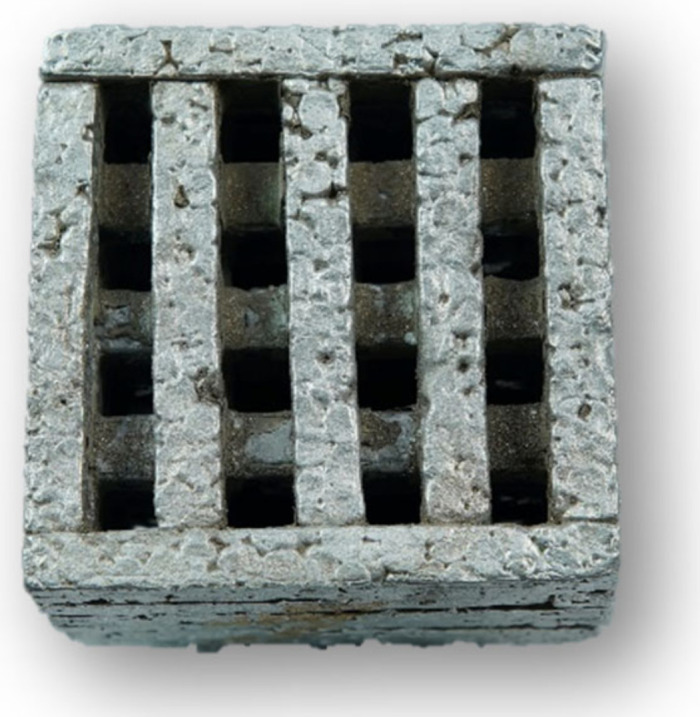
Detail of the casting after removal of the gating system.

**Table 1 materials-14-06989-t001:** Overview of foundry technologies for the production of metallic foams.

Technology	Moulding Material	Pattern	Material of Precursors/Preform	Casting Material
Infiltration of liquid metal into the cavity filled with precursors	Sand mould	Permanent pattern	Ceramic	Al-Si, cast iron
			Croning technology	Al-Si, Cu-Sn, cast iron
			Furan no-bake	Al-Si, Cu-Sn, cast iron
			PUR Cold Box	Al-Si, Cu-Sn, cast iron
			Salt	Al-Si, Cu-Sn
Two-stage investment casting process	Plaster	Polyurethane evaporable pattern	–	Al-Si, Cu-Sn
Infiltration of liquid metal into a cavity filled with a preform	Sand mould	Permanent pattern	PUR Cold Box	Al-Si, Cu-Sn, cast iron
Technology using a disposable evaporable pattern	Silica sand	Polystyrene evaporable pattern	–	Al-Si

**Table 2 materials-14-06989-t002:** Basic parameters of foams used for the production of patterns.

Name	Number of Pores (PPI)	Density (kg∙m^−3^)	Cell Diameter (µm)
Bulpren S 28190	30	23–27	1650–2150
Bulpren S 28280	20	23–27	2300–3300
Bulpren S 32450 ^1^	10	27–31	3800–5200

^1^ This foam has proven to be the best in terms of fluidity.

**Table 3 materials-14-06989-t003:** summary of results—achieved parameters of the resulting metallic foam castings.

Technology		Pore Content (%)	Pore Ø (mm)	The Main Advantage of the Technology
Infiltration of liquid metal into the cavity filled with precursors	Ceramic	50–60	8–16	Available material
	Croning technology	60–70	10–30	Use of waste material
	Furan no-bake	50–60	15–20	Good collapsibility
	PUR Cold Box	50–60	10	Precursors of the same shape and size
	Salt	50–60	18–22	Solubility of precursors in water
Two-stage investment casting process		Up to 97	4–5	Achieving high pore content
Infiltration of liquid metal into a cavity filled with a preform		45–55	10	Regular distribution of internal cavities
Technology using a disposable evaporable pattern		50	6	Regular distribution of internal cavities

## Data Availability

Not applicable.

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
