# Peer review of "Preparation of Cast Metallic Foams with Irregular and Regular Inner Structure"

_materials, 2021, doi:10.3390/ma14226989_

Round 1

Reviewer 1 Report

This paper gives foundry methods for the production of metal foams. Foundry production methods offer the possibility of using common procedures and standard materials. So the content is interesting. However, the authors did not show the advantages of this method very clearly compared with other methods. By this method, what are the ranges of porosity,  pore density,  pore diameter? Does the hollow problem of skeleton can be soloved?  Finally, the authors can add figures to give more details about the procedures of casting.

Reviewer 2 Report

Thought the title of present paper is somewhat interesting, it is not written well. Very poorly constructed sentences. Hence, I am not recommending this article in present form. I have observed the following points which may improve the article if the authors are ready to revise it.

1) Authors names, department, and affiliation are available in the first page of article. Hence, it is not necessary to incorporate in the abstract (Page no 1, line no 16). Further, the authors have used “thanks” which is not appropriate in a scientific article

2) Most of the sentences in the abstract are un general one. Here, the authors have to mentioned their focussed points in the article in a short form.

3) Out of 19 references, only 3 are from past 5 years and the remaining are old. The authors are asked to refer latest articles, and accordingly, the manuscript has to be re-written in the appropriate places

4) What are the specific applications for metallic foams? It has to be incorporated in the introduction

5)  Several literatures related to the development of metallic foams are to be incorporated in the introduction part (at least 10 to 15). Elaborate discussions of each article is to be addressed in introduction

6) The main objectives of the present work is to be added at the end of introduction part

7) In Materials and Methods, for ex. “The porosity can vary from 30 to 97%, depending on the production method and the 79 used material”. This is like a general statement. The authors have to study deeply the corresponding literature and write the major outcome. Similarly, in the coming citations

8) “Porous 87 metal materials can be made of metal vapor, liquid metal, powdered metal or ionized 88 metal”. Here, the name and meaning of “metal vapour” is wrong. The exact processing technique is to be incorporated

9) Figures 1 and 2 are the authors original? Or if it is used from anywhere mean, it has to be cited with the appropriate references. In similar manner for all other Figures.

10) There are no scale bar in Figures 1 and 2

11)  When I read the article, I couldn’t able to find any useful information’s. Language of the present article is very poor. Hence, my recommendation is to re-write the article with more scientific informations

Reviewer 3 Report

In this manuscript, the authors carefully presented foundry-based processes for making cellular metals (foams) that were implemented at the authors’ institution. The studied approaches are not new; they were reported before in many reviews, including those that were referred to in the manuscript. The authors described technological details, such as the commercial precursor compositions tried, mould materials, pattern materials, the way of placing precursors into the mould cavity, etc. At the same time, there are no any dependencies on the process parameters, data on properties, process limitations, etc. So, this is not a typical scientific paper, and it is difficult to understand what is the key novelty as compared to the previously described processes.

That is why, the suggestions will be rather general than particular. I think, an additional table (or section) should be added to clearly summarize advantages of the studied approaches and explain what is the most prospective product’s application area where the other widespread methods do not meet the required combination of properties.

The big advantage of the reported processes is that the highly-porous cellular metals with open-cell structures were made by the gravity die casting.

In this respect, the readers would be very interested to see a quantitative description of the processes (mould temperatures, melt temperatures, pouring times, surface treatment of fine channels to improve the wetting, and so on). Investment casting with PU foam pattern is usually associated with the creation of significant pressure drops to fill channels with the diameter of << 1 mm corresponding to the PU foam ligament size. The readers will like to understand how the problem was eliminated in the presented processes.

COMMENTS:

1). Abstract, p.1, line 23

You wrote about metal foams as a class of materials: “ ….. the application potential of these materials is limited by the economic demands of their production technologies…”

It’s difficult to believe that your processes are economically competitive as compared to conventional foaming technologies (using propeller, by blowing). So, the competitiveness area should be defined.

2). You wrote, p.1, line 25: “..we designed, verified and optimized 4 main methods of metal foam production by the foundry technology   ”

In fact, no data on the optimized process parameters is available in the paper.

3). Since line 74:

“Materials and methods” section looks like a continuation of the state-of-the-art analysis (Introduction). Please, rename the item or merge. The methodologies are actually explained in the “Results” section in the respective sub-sections

4). Line 123, you wrote: “The aim was to quantify the basic parameters of the technology of metal foam production by casting into conventional foundry moulds on the basis of laboratory and semi-industrial tests.”

In fact, the authors did not report on the quantification results.

5). Line 151, you wrote: “The term precursors means spherical particles of organic or inorganic material”.

You’ll hardly find an organic precursor material that can survive at the pouring temperatures in your process.

6). Line 342, you wrote: “… surface treatment of a polymer pattern is the procedure with the application of one layer of paint - the patterns treated in this way led to an increase in the fluidity of the metal into the complex cavity of the mould.”

Could you, please, provide more details on this step: what paints were used and how they effected the fluidity (or, most probably, wetting).

7). Line 571, you wrote: “Foundry technologies could offer an economical, time-saving and in many cases ecological possibility of producing metal foams….”.

My opinion about economy was given above.

Ecology is an another discussing issue in your processes, even compared to foaming processes. For example, leachng with the nitric acid solution is applied (line 375). Also, in lines 439-445, we find the statement: “This core production technology is based on a two-component binder system (benzyletherpolyol and diphenylmethane-4-4-diisocyanate) and represents a suitable process for the production of such a geometrically blowing the core with a gaseous tertiary amine which acts as a catalyst to form a polyurethane resin. “ Again, regarding the evaporable pattern that is finally decomposed by heating (line 486): “The pattern is made of expanded polystyrene (EPS), assembled including the gating system by cutting, gluing and grinding”.

The ecology seems not to be among the strongest advantages of your processes.

8). Line 592, you wrote: “The course of the experiments made it possible to determine the optimal metal fluiditity conditions and pouring conditions for this type of castings with a very small metal fiber thickness (0.19 - 0.40 594 mm).”

This is what the readers would be interested to understand first of all in your research. Unfortunately, there is no a clear explanation in the manuscript.

Round 2

Reviewer 1 Report

The paper can be accepted in its current form.

Reviewer 2 Report

The authors have worked based on raised comments, and hence, I am recommending to consider for its publications

Reviewer 3 Report

You improved the manuscript, changed its structure, and reached an acceptable form. At the same time, the readers would be mostly interested in receiving more details of the process optimisation.